# Linking Fibrotic Remodeling and Ultrastructural Alterations of Alveolar Epithelial Cells after Deletion of *Nedd4-2*

**DOI:** 10.3390/ijms22147607

**Published:** 2021-07-16

**Authors:** Theresa A. Engelmann, Lars Knudsen, Dominik H. W. Leitz, Julia Duerr, Michael F. Beers, Marcus A. Mall, Matthias Ochs

**Affiliations:** 1Institute for Forensic Medicine, Hannover Medical School, 30625 Hannover, Germany; engelmann.theresa@mh-hannover.de; 2Institute of Functional and Applied Anatomy, Hannover Medical School, 30625 Hannover, Germany; 3Biomedical Research in Endstage and Obstructive Lung Disease Hannover (BREATH), Member of the German Center for Lung Research (DZL), 30625 Hannover, Germany; 4Department of Pediatric Respiratory Medicine, Immunology and Critical Care Medicine, Charité-Universitätsmedizin Berlin, Corporate Member of Freie Universität Berlin and Humboldt-Universität zu Berlin, Augustenburger Platz 1, 13353 Berlin, Germany; dominik.leitz@charite.de (D.H.W.L.); julia.duerr@charite.de (J.D.); marcus.mall@charite.de (M.A.M.); 5Department of Translational Pulmonology, Translational Lung Research Center (TLRC), Member of the German Center for Lung Research (DZL), University of Heidelberg, Im Neuenheimer Feld 156, 69120 Heidelberg, Germany; 6German Center for Lung Research (DZL), Associated Partner, Augustenburger Platz 1, 13353 Berlin, Germany; matthias.ochs@charite.de; 7Pulmonary, Allergy, and Critical Care Division, Department of Medicine, Perelman School of Medicine, University of Pennsylvania, Philadelphia, PA 19104, USA; mfbeers@pennmedicine.upenn.edu; 8PENN-CHOP Lung Biology Institute, Perelman School of Medicine, University of Pennsylvania, Philadelphia, PA 19104, USA; 9Berlin Institute of Health at Charité, Universitätsmedizin Berlin, Charitéplatz 1, 10117 Berlin, Germany; 10Institute of Functional Anatomy, Charité, Universitätsmedizin Berlin, 10117 Berlin, Germany

**Keywords:** *Nedd4-2*, pulmonary fibrosis, regeneration, alveolar epithelium, alveolar epithelial type 2 cells, surfactant, stereology

## Abstract

Our previous study showed that in adult mice, conditional *Nedd4-2*-deficiency in club and alveolar epithelial type II (AE2) cells results in impaired mucociliary clearance, accumulation of *Muc5b* and progressive, terminal pulmonary fibrosis within 16 weeks. In the present study, we investigated ultrastructural alterations of the alveolar epithelium in relation to interstitial remodeling in alveolar septa as a function of disease progression. Two, eight and twelve weeks after induction of *Nedd4-2* knockout, lungs were fixed and subjected to design-based stereological investigation at the light and electron microscopic level. Quantitative data did not show any abnormalities until 8 weeks compared to controls. At 12 weeks, however, volume of septal wall tissue increased while volume of acinar airspace and alveolar surface area significantly decreased. Volume and surface area of alveolar epithelial type I cells were reduced, which could not be compensated by a corresponding increase of AE2 cells. The volume of collagen fibrils in septal walls increased and was linked with an increase in blood–gas barrier thickness. A high correlation between parameters reflecting interstitial remodeling and abnormal AE2 cell ultrastructure could be established. Taken together, abnormal regeneration of the alveolar epithelium is correlated with interstitial septal wall remodeling.

## 1. Introduction

Idiopathic pulmonary fibrosis (IPF) is a progressively scarring lung disease with poor prognosis [1,2]. The etiology of this disease is complex and poorly understood and appears to be multifactorial, including both genetic and environmental factors as well as aging [3]. Genome-wide association studies identified variants of genes which increase the risk for IPF and are related to host defense (e.g., gain-of-function variant of *MUC5B*), telomere maintenance and finally, cell–cell contacts responsible for epithelial barrier functions [4]. Regarding environmental factors, air pollution including cigarette smoking is linked with an increased risk. In general, the disease has an insidious onset with a progressive fibrotic remodeling in baso-dorsal, subpleural lung regions, which are physiologically subject to largest deformation within the lung and therefore to parenchymal strain during spontaneous breathing. A recent study based on contrast enhanced dual energy computed tomography demonstrated that increased regional ventilation was a predictive parameter for future decline in lung function, and regions which were subject to higher regional ventilation (deformation) were prone to fibrotic remodeling in follow-up, showing that mechanical stress is of relevance for disease progression [5,6,7].

There is good evidence that the fibrotic remodeling is a response upon chronic injury of the alveolar epithelium combined with abnormal regeneration [3]. In healthy lung parenchyma, the alveolar epithelium is a mosaic composed of 2 different cell types. While the alveolar epithelial type I (AE1) cells are responsible for minimizing the blood–gas barrier with squamous cell extensions, thereby covering approximately 95% of the surface area, the cuboidal alveolar epithelial type II (AE2) cells are in charge of surfactant homeostasis and epithelial cell regeneration [8]. In the context of the pathogenesis of IPF, the AE2 cells have been assigned a central role [9,10]. Dying AE2 cells have been observed in IPF in remodeled [3,11], but also in not (yet) remodeled regions [12], and at the ultrastructural level, cell loss with denudation of the epithelial basal lamina was a quite common finding. In human IPF and animal models, apoptosis of AE2 cells has been linked with a diversity of abnormalities, including increased endoplasmic reticulum (ER) stress and autophagy, mitochondrial dysfunction and accelerated senescence [10]. Hence, the combination of ongoing lung injury with impaired regeneration, e.g., a failure to repopulate the denuded basal lamina by dividing AE2 cells, represents a trigger for pulmonary fibrosis [13,14].

The mechanisms which lead to the ongoing lung injury in IPF are poorly understood. Mucociliary clearance is a very important defense mechanism of the respiratory system and is impaired in IPF. Recent studies have provided evidence that reduced mucociliary clearance is linked with progressive pulmonary fibrosis in mice so that the failure of clearance of, e.g., inhaled particles, is a potential mechanism of injury of the epithelium [15,16]. Duerr et al. induced a conditional deficiency of *Nedd4-2* (a gene also known as *Nedd4L*) in club cells and AE2 cells in adult mice. *Nedd4-2* (neuronal precursor cell expressed developmentally downregulated protein 4) is an E3 ubiquitin protein ligase. It regulates cellular trafficking, including endocytosis and lysosomal degradation of ion channels [17]. In this regard, it reduces the resorption of sodium and water in the airways by removal of epithelial sodium channels (ENaC) from the apical plasma membrane. Hence, lack of *Nedd4-2* in club cells and AE2 cells results in an increased absorption of sodium and water, which reduces the efficiency of mucociliary clearance [18,19]. Congenital *Nedd4-2* deficiency was linked with severe inflammation and acute lung injury of newborn mice with high neonatal mortality [20]. However, following the time course of pathophysiological events after conditional knockout of *Nedd4-2* in adult mice illustrated progressive airway remodeling with an increased density of goblet cells in distal conducting airways, accumulation of the gel-forming mucin *Muc5b* and fibrosis in peripheral lung regions. The disease progression was characterized by a dramatic acceleration between 8 and 12 weeks of conditional *Nedd4-2* depletion [15].

However, *Nedd4-2* not only regulates cellular degradation of ENaC, but also terminates TGF-β1 signaling by ubiquitination of linker phosphorylated Smad2/Smad3 [15,21] and is necessary for trafficking of pro-surfactant protein C to the storing organelle within AE2 cells, the so-called lamellar body [22,23]. Hence, lack of *Nedd4-2* appears to create a pro-fibrotic environment in lung parenchyma via several pathways, also including an increased TGF-β1 signaling [15]. Compared to many other animal models of pulmonary fibrosis [24,25], the conditional knockout of *Nedd4-2* in adult mice shares several important features with IPF, and these include the insidious development of the disease, the distribution and pattern of lesions (e.g., honeycombing) and the accumulation of *Muc5b* in distal parts of the lung [26], just to name a few. Of interest is also the fact that previous studies found reduced *NEDD4L* expression at the RNA and protein levels in IPF lungs, observations which emphasize the relevance of this model in the context of IPF [15,27].

In view of the fact that the alveolar epithelium has been attributed a central pathophysiological role in IPF, this study focuses on the changes occurring in the AE2 cells during disease progression. We aimed to investigate the following questions: (1) Does the conditional deletion of *Nedd4-2* have a direct effect on the ultrastructure of the alveolar epithelium, including the intracellular surfactant system? (2) Do ultrastructural abnormalities of the alveolar epithelium and the intracellular surfactant pool correlate with the interstitial remodeling? (3) In this animal model of progressive pulmonary fibrosis, is there any evidence for insufficient epithelial regeneration?

Two, eight and twelve weeks after induction of *Nedd4-2* deficiency, lungs were fixed for quantitative morphology by means of design-based stereology at the light and electron microscopic level. This study was based on our previous findings that demonstrated the absence of light microscopical abnormalities 2 weeks post-induction of *Nedd4-2* deficiency but a discrete thickening of septal walls at 8 weeks and fibrosis 12 weeks after induction of *Nedd4-2* deletion [15]. Hence, these timepoints reflected different stages of disease progression. At the light microscopic level, the volume of acinar airspaces and septal wall tissue per lung was determined. In a second step, the components of the interalveolar septa were quantified at the ultrastructural level, including, among others, the volumes of the different alveolar epithelial cell populations, and extracellular matrix components such as collagen fibrils and interstitial cells. In addition, the surface area of the alveolar epithelium covered by air was quantified as well as the surface areas of the alveolar epithelial basal lamina covered by different epithelial cell populations were determined. Finally, alterations of the parameters characterizing the alveolar epithelium were correlated to parameters quantifying interstitial remodeling within interalveolar septa.

## 2. Results

### 2.1. Progressive Fibrotic Remodeling in Nedd4-2-Deficient Lungs

Figure 1 illustrates representative light microscopic images of control (ctrl) and *Nedd4-2*-deficient lungs (*Nedd4-2*^−/−^) after 2, 8 and 12 weeks of doxycycline induction. After 2 and 8 weeks, there were no apparent differences between controls and *Nedd4-2*-deficient lungs. After 12 weeks, septal walls appeared thicker and inflammatory infiltrates could be observed in *Nedd4-2*-deficient lungs. Some areas of these lungs were consolidated and destroyed so that the typical acinar components such as airspaces and interalveolar septa were no longer definable. These consolidated areas were therefore not considered as reference spaces for our stereological investigations.

A design-based stereological analysis was performed, and data with regard to alveolar microarchitecture are provided in Table 1. In all study groups, both female and male mice were included. In particular, in the *Nedd4-2* knockout groups, the numbers of female and male mice were balanced. There was a significant increase in the lung volume (V(lung)) with time, which was most prominent between 14 and 18 weeks of age (corresponding to 8 and 12 weeks of the study) in both control and *Nedd4-2*-deficient lungs. No significant differences between control and *Nedd4-2*-deficient lungs were observed regarding volume of parenchyma (V(par,lung)), but there was a significant time effect in both genotypes, in particular between 8 and 12 weeks. The volume of septal walls increased significantly in *Nedd4-2*-deficient lungs between 8 and 12 weeks with contemporaneously reduced volume of ductal and alveolar airspace (V(air,par)).

### 2.2. Composition of the Interalveolar Septa

Electron microscopy was used to analyze the composition of the interalveolar septal walls. Figure 2 illustrates representative electron microscopic images of control (ctrl) and *Nedd4-2*-deficient lungs after 2, 8 and 12 weeks. After 2 and 8 weeks, there were no apparent differences of the ultrastructure between controls and *Nedd4-2*-deficient lungs. After 12 weeks, a fibrotic interstitial remodeling within interalveolar septal walls with an increase of collagen fibrils, interstitial cells and AE2 cells could be observed. Moreover, intermediate cells which shared features of AE1 and AE2 cells were found as a third type of alveolar epithelial cells.

Based on these qualitative observations, we quantified the composition of the interalveolar septal walls by using design-based stereology. These data are summarized in Table 2 and shown in part in Figure 3. Within the first 8 weeks after induction of *Nedd4-2* deficiency, there was no fibrotic interstitial remodeling within interalveolar septal walls, such as increased collagen deposition or thickening of the blood–gas barrier. Between 8 and 12 weeks, the arithmetic mean thickness of the blood–gas barrier of *Nedd4-2*-deficient mice increased significantly (Figure 3F). The components of the blood–gas barrier were analyzed and showed a significant increase of the volume of AE2 cells (Figure 3B) and the volume of interstitial cells (Figure 3C). This was accompanied by a simultaneous, significant decrease of the volume of AE1 cells (Figure 3A). The interstitium consists of interstitial cells (IC) and extracellular matrix (ecm). This extracellular matrix can be differentiated into collagen fibrils (col) and extracellular matrix other than collagen fibrils (_r_ecm). The volumes of these components were quantified and showed a significant increase of collagen fibrils between 8 and 12 weeks of *Nedd4-2* deficiency, with contemporaneous stability of extracellular matrix other than collagen fibrils (Figure 3D,F). The controls showed stable parameters.

In addition, there was an increase of the absolute volume of lamellar bodies per lung and the volume-weighted mean volume of the lamellar bodies between 8 and 12 weeks of *Nedd4-2* deficiency compared to the control group (Figure 3G,H).

We did not find gender differences in the composition of the interalveolar septa in healthy controls. However, after 12 weeks of *Nedd4-2* deficiency, the volume of AE2 cells (42.2 (11.3) mm³ vs. 17.1 (3.2) mm³, *p* = 0.021) and the volume of lamellar bodies per lung (8.38 (1.64) mm³ vs. 3.56 (1.11) mm³, *p* = 0.014) were larger in males compared to females. No gender effects were found with regard to interstitial remodeling.

### 2.3. Alveolar Epithelial Cell Injury

In areas of increased fibrotic interstitial remodeling within interalveolar septal walls, a third type of alveolar epithelial cells could be localized. These intermediate cells represent characteristics of AE1 and AE2 cells and were only observed after 12 weeks of *Nedd4-2* deficiency (Figure 4A). At the electron microscopic level, the surface fraction of the basal lamina covered by AE1 cells, AE2 cells and intermediate cells in controls and *Nedd4-2*-deficient lungs were analyzed. *Nedd4-2*-deficient lungs showed a significantly increased surface area of the epithelial basal lamina covered by AE2 cells and intermediate cells, with contemporaneously reduced surface area of the epithelial basal lamina covered by AE1 cells. The controls showed stable parameters (Figure 4B). A sub-group analysis of lungs investigated 12 weeks after induction of *Nedd4-2* knockout did not find differences in the surface area of the basal lamina covered by AE1 cells or intermediate cells between females and males. Nevertheless, a trend can be described for an increased surface area of the basal lamina covered by AE2 cells in males compared to females (120 (38) cm² vs. 86 (17) cm²; *p* = 0.096).

The surface areas of alveolar epithelial cells covered by air were stable during the first 8 weeks (Table 2). A reduction of surface area by approximately 1/3 compared to the control group (Table 3) was observed at 12 weeks of *Nedd4-2* deficiency. Additionally, there was a significant decrease of surface area of AE1 cells covered by air and an increase of surface area of intermediate cells covered by air. Nevertheless, the *Nedd4-2*-deficient group demonstrated a trend of an increase in the surface area of AE2 cells covered by air between 8 and 12 weeks (Table 3).

The surface area of endothelial cells after 12 weeks of *Nedd4-2* deficiency compared to the controls appeared roughly stable (Table 3).

### 2.4. Correlation Analyses

We characterized relationships between different parameters of the ultrastructure of the blood–gas barrier of *Nedd4-2*-deficient (*Nedd4-2*^−/−^) lungs after 12 weeks using the Spearman correlation (Figure 5). The thickness of the blood–gas barrier demonstrated a highly significant correlation with the fraction of surface area of basal lamina covered by AE2 cells and the volume-weighted mean volume of lamellar bodies (Figure 5A,B). The volume of collagen fibrils showed no significant correlation with the fraction of surface area of basal lamina covered by AE2 cells (Figure 5C), but there was a highly significant correlation between the volume of collagen fibrils and the volume-weighted mean volume of lamellar bodies (Figure 5D). Accordingly, the volume of interstitial cells demonstrated a positive significant correlation with the fraction of surface area of basal lamina covered by AE2 cells and a positive, highly significant correlation with the volume-weighted mean volume of lamellar bodies (Figure 5E,F).

Taken together, after 12 weeks, the *Nedd4-2*-deficient lungs showed a significant correlation between the fraction of the surface area covered by AE2 cells and the parameters to quantify the interstitial fibrotic remodeling, such as the arithmetic mean thickness of the blood–gas barrier, the volume of collagen fibrils and the volume of interstitial cells. Additionally, there was a highly significant correlation between volume-weighted mean volume of lamellar bodies, as a parameter of the intracellular surfactant storage pool, and parameters of fibrosis.

## 3. Discussion

*Nedd4-2* deficiency has been shown to trigger fibrotic remodeling within the lung parenchyma by chronic injury of the epithelium, among others due to reduced mucociliary clearance, a mechanism which has been suggested to be of relevance for formation of microscopic honeycomb cysts in IPF [15,28]. In this regard, alterations of the composition of the respiratory epithelium of the purely conductive airways have been described. However, in the conditional *Nedd4-2* knockout model, nothing is known about how changes in the ultrastructure of the alveolar epithelium are linked with the interstitial fibrotic remodeling within the interalveolar septal walls. Hence, we applied design-based stereology, the current gold standard for quantitative morphology at the light and electron microscopic level [29], to characterize structural abnormalities during the time course of disease progression. We focused on parameters of interstitial fibrosis, and quantified the arithmetic mean thickness of the blood–gas barrier, the volume of collagen fibrils and interstitial cells within interalveolar septa, and related these parameters to the composition of the alveolar epithelium, which is normally formed by a mosaic of AE1 and AE2 cells.

Our data confirm previous findings [15] that within the first 8 weeks after induction of *Nedd4-2* deficiency in adult mice, no obvious abnormalities in acinar microarchitecture can be found (Figure 1, Table 1). In the present study, there was no increased collagen deposition and thickening of the blood–gas barrier, and thus no fibrotic interstitial remodeling within interalveolar septal walls (Table 2, Figure 3). Although the *Nedd4-2* deficiency was specific for club cells and AE2 cells [15], the ultrastructure of the alveolar epithelium was maintained and the volumes of AE1 and AE2 cells did not differ between study groups. AE2 cells play a central role in the surfactant metabolism [30]. Surfactant, a complex mixture of phospholipids and surfactant proteins A–D, is synthesized, stored and released by AE2 cells. Storage within AE2 cells takes place in a unique organelle, termed lamellar body, in which phospholipid membranes are onion-like and densely packed and coated by a limiting membrane. Intracellular surfactant is released into the airspace, where it is essential for reducing surface tension at the alveolar air–liquid interface at end-expiration, so that it stabilizes the surface area available for gas-exchange throughout the respiratory cycle and reduces the inspiratory work load. Until week 8 of induction of *Nedd4-2* deficiency, the intracellular surfactant pool, defined as the total amount of the lamellar bodies in AE2 cells [31], did not differ from age-matched controls, although the trafficking of prosurfactant protein C towards the lamellar bodies depends on *Nedd4-2*. Accordingly, in our present study, neither the absolute volume of lamellar bodies in the lung nor the volume-weighted mean volume of lamellar bodies differed between *Nedd4-2*-deficient and age-matched control groups. Hence, we can conclude that the knockout as such does not result in ultrastructural abnormalities of AE2 cells.

However, a considerable dynamic in disease progression could be detected between 8 and 12 weeks of *Nedd4-2* deficiency. Within this brief period, a small fraction of 5–6% of the lung volume was completely destroyed and consolidated so that the typical acinar components such as airspaces and interalveolar septa were not definable any more. These consolidated areas were therefore not considered as reference space for our stereological investigation, which focuses on the alterations of the acinar microarchitecture induced by conditional *Nedd4-2* deletion. Within the interalveolar septa, the volumes of collagen fibrils and interstitial cells were approximately triplicated in *Nedd4-2* knockout mice but remained stable in the controls (Table 2). This was associated with a considerable increase in the thickness of the blood–gas barrier and loss of surface area of alveolar epithelial cells (Table 2 and Table 3, Figure 3), both of them being important structural characteristics for an efficient gas-exchange [6]. The dimension of the increase in the volume of collagen fibrils in septal walls was comparable to other models of pulmonary fibrosis. Two weeks after induction of fibrosis with amiodarone or adenoviral vector-mediated transfer of active TGF-β1 gene to epithelial cells, a 3- to 5-fold increase in the volume of collagen fibrils could be detected [32,33,34].

The development of interstitial fibrosis 12 weeks after induction of *Nedd4-2* deficiency was correlated with abnormalities found in the alveolar epithelium. There was a loss of AE1 cells, as indicated by a reduction in the total volume of this cell population within the lung (Figure 3A), escorted by a reduced surface area of the epithelial basal lamina covered by AE1 cells. Instead, the absolute volume of AE2 cells (Figure 3B) as well as the surface area of the basal lamina covered by AE2 cells significantly increased (Table 3, Figure 4). However, the loss of basal lamina surface area covered by AE1 cells could only be compensated in part by an increase in AE2 cells, so that compared to age-matched control lungs, the surface area of the basal lamina decreased by approximately 360 cm². Taken together, although there was a shift from AE1 cells to AE2 cells, the loss of basal lamina surface area covered by AE1 cells could not be compensated entirely. These observations can be interpreted as an insufficient regenerative response of the alveolar epithelium to a chronic injury affecting the blood–gas barrier between weeks 8 and 12 after depletion of *Nedd4-2,* but not before week 8, since the surface area of alveolar epithelium covered by air was stable and did not differ between the genotypes during the first 8 weeks.

In the control group, the surface area increased between 8 and 12 weeks, in parallel with the lung volume, and this is likely the consequence of ongoing lung development in adult mice, as reported before [35]. The loss of surface area of approximately 1/3 in *Nedd4-2*^−/−^ compared to the control group (Table 3) might be explained by disturbed lung development. However, we did not find differences in lung volumes, and in the *Nedd4-2*^−/−^, there was a shift of volume fractions: fractions of acinar airspaces were reduced, while those of interalveolar septa increased. In concert with a quite moderate consolidation amounting to 5–6% of lung volume (Table 1), the reduced surface area of air covered epithelium can also be explained by a mechanism called collapse induration. Injury of the blood–gas barrier is linked with instability of alveolar airspaces so that these partially or completely collapse. With inefficient regeneration, alveoli remain collapsed, their entrances are overgrown by epithelial cells and finally, the collapsed alveoli are engulfed in fibrosis. In essence, a proportionally small increase in destroyed and fibrotic lung parenchyma results in a considerable loss of surface area. The mechanism of collapse induration has been described at the ultrastructural level in animal models of lung injury and fibrosis [34,36], but also in human lung injury due to COVID-19 [37] or idiopathic interstitial pneumonia including IPF [3,38,39]. The mechanism of collapse induration can explain the considerable loss of alveolar surface area by approximately 2/3 found in IPF without much increase in tissue volume [40], a feature which the conditional *Nedd4-2* deficiency model partially reproduces.

In the current animal study, however, we did not observe direct evidence for epithelial cell injury, such as denudations of the epithelial basal lamina, which is a typical ultrastructural finding in human IPF [3]. In view of the fact that the *Nedd4-2*^−/−^ is a chronic model with a very slow progress, it might be possible that the injury is too subtle to be detected in those regions in which the acinar microarchitecture could still be analyzed. However, even in models of acute lung injury, it has been demonstrated that only a few percent (3%) of the alveolar epithelium are destroyed and the basal lamina denuded [41,42], so in the present study, it is likely that we simply missed areas demonstrating this rare pathological lesion.

Lung injury as such has been shown to induce a local hyperplasia of AE2 cells [34,36,43], a finding that we also observed in our model, in particular in areas of increased interstitial fibrotic remodeling (Figure 2). Moreover, a third type of alveolar epithelial cell which shared features of AE2 and AE1 cells could be localized in areas of increased deposition of collagen fibrils, and was termed intermediate cell (Figure 4). Epithelial cells with comparable ultrastructural features have been demonstrated in human lungs suffering from acute respiratory distress syndrome (ARDS), including prolonged ARDS due to COVID-19 [37,44] as well as in human IPF samples [3]. Similar to prolonged fibrosing ARDS or IPF, the alterations found in the alveolar epithelium in the present study, e.g., the shift from AE1 to AE2 cells or the occurrence of intermediate cells, are suggestive of abnormal epithelial regeneration after injury.

In acute lung injury resulting in regeneration, AE2 cell proliferation is increased and AE2 cells transdifferentiate to AE1 cells [45,46,47]. The transdifferentiation from AE2 to AE1 cells is associated with a loss of AE2 cell characteristics, such as the ability to produce and store surfactant in lamellar bodies or to enter the cell cycle. In an animal model of acute lung injury, the transdifferentiation has been shown to take place between days 7 and 16 after the insult [41].

In the present study, it appeared that regeneration and the process of transdifferentiation failed. The AE2 cells were increased in number and/or individual size, contained a larger volume of lamellar bodies per lung and the volume-weighted mean volume of lamellar bodies was increased, which can result from both increased mean size and increased size variability of lamellar bodies. A proportion of cells covering interalveolar septal walls were characterized as intermediate epithelial cells, describing, based on ultrastructural criteria, a state between AE2 and AE1.

The mechanisms of transdifferentiation are not well-understood, but require a temporally and spatially well-orchestrated up- and down-regulation of signaling pathways, thereby promoting proliferation and arrest of proliferation of AE2 cells followed by transdifferentiation [48]. A recent study using single-cell RNA-sequencing identified different types of regenerating alveolar epithelial cells during acute lung injury based on the TGF-β1 signaling. In particular, in cells characterized as intermediate between AE2 and AE1, the TGF-β1 signaling was increased, while in those cells differentiating towards AE1, the TGF-β1 signaling was downregulated again [49]. Based on these observations, it has been suggested that persisting TGF-β1 signaling stops transdifferentiation from AE2 to AE1 after injury. In line with these observations, Jiang and co-workers identified epithelial cells which were fixed in an intermediate state between AE2 and AE1 cells in IPF samples and in the bleomycin model of lung injury and fibrosis [50].

In our model, *Nedd4-2* deficiency resulted in increased phospho-Smad2/Smad3 levels in lung parenchyma, and therefore persisting intracellular TGF-β1 signaling [15]. In addition, the TGF-β1 level was increased after 12 weeks but not 2 or 8 weeks after induction of *Nedd4-2* deletion [15]. This increase correlated with the occurrences of intermediate alveolar epithelial cells, AE2 cell hyperplasia and the increase of the fraction of the surface area of the epithelial basal lamina covered by AE2 at the expense of AE1 cells. Hence, increased TGF-β1 signaling might at least in part have mediated the alterations we observed with regard to the alveolar epithelium in our animal model.

A detailed analysis of the stereological data after 12 weeks of induction of *Nedd4-2* depletion demonstrated a strong correlation between the fraction of the surface area covered by AE2 cells and the parameters used to quantify the interstitial remodeling. The same could be observed with regard to abnormalities of the intracellular surfactant: the volume-weighted mean volume of lamellar bodies highly correlated with fibrosis (Figure 5). These findings reproduce observations from previous studies including the amiodarone, TGF-β1 and bleomycin model, which are, unlike the current model, rather models of acute lung injury and fibrosis [33,36,51,52,53]. In human IPF, AE2 cells are also subject to cellular stress, and ultrastructural abnormalities of the intracellular surfactant represent typical findings [3,54]. Kawanami et al. performed an electron microscopic study of the epithelium in IPF and observed that the hypertrophy and hyperplasia of AE2 cells correlated with the degree of deposition of collagen fibrils in discrete to moderate fibrosis, while in areas with severe fibrosis, different cuboidal epithelial cells were observed.

IPF and the conditional *Nedd4-2* knockout model share the feature that the gel-forming *Muc5b* is upregulated in epithelial cells within the lung periphery, including AE2 cells and other cuboidal cells [15,16,26,55]. In the background of *Nedd4-2* deficiency, this increased expression of *Muc5b* aggravates mucus concentration and therefore mucociliary dysfunction, a potential mechanism for fibrotic remodeling [28]. It has been shown that impaired mucus clearance resulting in an excess of mucus in the conducting airways is sufficient to trigger chronic inflammation and related lung diseases in the absence of a second hit such as an infection [56]. In addition, recent studies demonstrated that the increased expression of *Muc5b* in epithelial cells of distal airways also induces cellular stress [57] so that the observed alterations of the intracellular surfactant might also reflect cellular stress, e.g., due to upregulated *Muc5b*.

Aside from stressing epithelial cells in lung parenchyma, the occurrence of *Muc5b* in distal airspaces also interferes with the function of the intra-alveolar surfactant. Nguyen et al. demonstrated that *Muc5b* increases surface tension in alveoli and concluded that the excess of mucins in the lung periphery is a potential mechanism for injury of the blood–gas barrier via mechanical stress, e.g., during mechanical ventilation [58,59,60]. Even during spontaneous breathing, high surface tension is sufficient to induce injury of alveolar epithelial cells [61]. In the context of the present study, high surface tension and related alveolar instability might explain the loss of epithelial surface area due to collapse induration, as discussed earlier. However, due to the airway instillation fixation of the lungs in our study, we were not able to investigate the effect of high surface tension on the acinar microarchitecture [62]. If high surface tension is of relevance, one would expect to observe loss of air due to alveolar collapse, in particular at low lung volumes in air-filled lungs [53], a process which has also been observed in IPF and precedes shrinkage and fibrotic remodeling [5].

In conclusion, in the model of *Nedd4-2* deficiency, the abnormalities of the alveolar epithelium are not a direct consequence of the lack of *Nedd4-2,* since the ultrastructure did not differ from the wild-type within the first 8 weeks. After 12 weeks of *Nedd4-2* knockout, however, interalveolar wall remodeling correlated highly with an abnormal composition of the alveolar epithelium as well as alterations of the intracellular surfactant pool. At which time point the abnormalities of the alveolar epithelium become apparent and whether these even precede the remodeling is not clear and would have required investigations of additional time points between week 8 and week 12 of *Nedd4-2* depletion. Our data suggest that there is an ineffective regeneration of the alveolar epithelium characterized by a loss of epithelial surface area for gas-exchange combined with the occurrence of intermediate epithelial cells, a decrease of AE1 and an increase in AE2 cell volume and surface area. Based on our previous study [15], these structural alterations correlate with *Muc5b* overexpression and increased TGF-β1 signaling in lung parenchyma. Aside from impaired mucociliary clearance which injures the epithelium, it is likely that *Muc5b* overexpression is involved in inducing harmful epithelial ER stress and high surface tension-related mechanical stress within alveoli, while increased TGF-β1 signaling has the potential to interfere with efficient regeneration.

## 4. Materials and Methods

### 4.1. Animal Model, Study Groups, Sampling and Embedding for Light and Electron Microscopy

All animal studies were approved by the animal welfare authority responsible for the University of Heidelberg (Regierungspräsidium Karlsruhe, Karlsruhe, Germany). Mice for conditional deletion of *Nedd4-2*^−/−^ were generated as previously described [15]. In brief, mice carrying loxP sites flanking exon 15 of *Nedd4-2* (*Nedd4-2^fl/fl^*) [63] were intercrossed with *CCSP-rtTA2^S^-M2* line 38 (*CCSP-rtTA2^S^-M2*) [64] and *LC1* mice [65,66] to achieve doxycycline-dependent deletion of *Nedd4-2* in club and AE2 cells.

For longitudinal assessment of pulmonary structural investigations, 4- to 6-week-old *Nedd4-2^fl/fl^/CCSP-rtTA2^S^-M2/LC1* mice and littermate controls were treated with doxycycline for 2, 8 and 12 weeks, as indicated: 6 control and 6 *Nedd4-2*^−/−^ lungs per group (2 weeks *Nedd4-2*^−/−^: N = 6 (females 3, males 3); 2 weeks control: N = 6 (females 4, males 2); 8 weeks *Nedd4-2*^−/−^: N = 6 (females 3, males 3); 8 weeks control: N = 6 (females 3, males 3); 12 weeks *Nedd4-2*^−/−^: N = 6 (females 3, males 3); 12 weeks control: N = 6 (females 4, males 2)) were fixed ex situ by airway instillation using a hydrostatic pressure of 25 cm H_2_O with 1.5% paraformaldehyde and 1.5% glutaraldehyde in 0.15 M HEPES buffer (pH 7.35). After the total lung volume (V(lung)) was determined based on the principle of Archimedes by fluid displacement [67], the whole lung was embedded in 4% agar and cut with a tissue slicer into slices of equal thickness from the apex to the base. A systematic uniform random sampling was performed according to standardized protocols to obtain an unbiased and representative set of tissue slices for light and electron microscopic investigations of the lung [68]. Tissue slices sampled for light microscopic evaluation were embedded in glycol methacrylate (Technovit 8100^®^, Haereus Kulzer, Wehrheim, Germany) to minimize tissue deformation [69]. The embedded tissue was cut into 1.5 μm thin sections and stained with toluidine blue. Based on established protocols [70], small tissue blocks for electron microscopy were embedded in epoxy resin (Epon^®^, Serva Electrophoresis GmbH, Heidelberg, Germany) and cut into 60 nm ultrathin sections.

### 4.2. Design-Based Stereology

The methodology of stereology was applied to determine structural changes in the lungs as a function of the duration of *Nedd4-2* deficiency after 2, 8 and 12 weeks. All methods for quantitative assessment of lung structure were performed according to the recommendations of the American Thoracic Society (ATS) and European Respiratory Society (ERS) [29]. At the light microscopic level, the newCAST software (Version 3.2, Visiopharm, Hoersholm, Denmark), with test systems such as test points and test lines, were used by the blinded investigator. To ensure precision, the unbiased test system was adapted in a way that leastways 100–200 counting events per parameter and lung were generated. First, the lung was investigated with a magnification of 5×, and test points were projected onto the sampled fields to differentiate between parenchymal (involved in gas-exchange) and non-parenchymal (conducting airways, larger vessels and perivascular connective tissue) components. The volume fractions of parameters were multiplied by the lung volume (reference space) in order to avoid falling into the “reference trap”, defined by Braendgaard and Gundersen [71]. Secondly, the parenchyma was differentiated at a higher magnification (20×) with test points into alveolar and ductal airspaces V(air,par) and septal tissue V(sep,par). Twelve weeks after induction of *Nedd4-2*^−/−^ deficiency, the consolidated, destroyed tissue areas were additionally determined V(cons,lung) [72,73]. For further evaluations at the electron microscopic level, the volume of the interalveolar septa V(sep,par) represented the reference space.

At the electron microscopic level, the composition of the interalveolar septa, including the arithmetic mean thickness of the blood–gas barrier and the surface area of the alveolar epithelium and endothelium, was estimated using single sections. An unbiased, representative set of images was sampled by means of systematic uniform random area sampling on mean 5 ultrathin sections per lung based on the y–x coordinates of the stage of the Morgagni transmission electron microscope (FEI, Eindhoven, The Netherlands) fitted out with a Valeta digital camera (Olympus Soft Imaging Solutions, Münster, Germany). Using a primary magnification of 8900×, the septal wall was imaged every 50 μm. The consolidated, destroyed tissue areas were excluded from the measurements at the electron microscopic level. Thereafter, the images were analyzed by using the Stepanizer online tool [74]. With the Stepanizer online tool (www.stepanizer.com), a test system for point counting and intersection counting was superimposed on the electron microscopic images. Point counting was used to determine volume fractions of AE1, AE2, the endothelium (Endo), interstitial cells (IC), capillary lumen (capl), collagen fibrils (defined by ultrastructural criteria) and residual extracellular matrix (_r_ecm) without collagen fibrils. In addition, in the 12 weeks *Nedd4- 2^−/−^* group, intermediate epithelial cells (defined by ultrastructural criteria) were determined within the interalveolar septa. Intersection counting was used to determine the surface area of the endothelium and epithelium covered by air. In addition, after 12 weeks of *Nedd4-2* deficiency, the surface area of the epithelial basal lamina covered by AE1, AE2 and intermediate cells was quantified [61]. Next, the total volume of AE2 cells within interalveolar septa represented the reference space [32]. Using a primary magnification of 7100×, the AE2 cells were imaged every 50 μm and the volume of the lamellar bodies was estimated by point counting. The point-sampled intercept method was applied to determine the volume-weighted mean volume of LB (ν_v_(LB)) [75]. The light and electron microscopic parameters are defined in Table 4 and Table 5.

### 4.3. Statistical Analyses

Generated data were tested for normality using the Kolmogorov–Smirnov test. To investigate the effects of the factors of genetics and time and the interaction between these factors, a two-way ANOVA followed by the post hoc Tukey test for multiple comparisons and adjustment of the *p*-level were used to take these influencing factors into account. For these statistical analyses, Sigma Plot Version 13.0 (Systat Software, Düsseldorf, Germany) was used. Generally, *p*-values below 0.05 were considered as statistically significant. Male and female lungs were investigated in this study. Twelve weeks after induction of *Nedd4-2* deficiency, subgroup analyses were performed between males and females (*n* = 2–4 per subgroup) to explore gender effects in this model using a *t*-test. Correlation analyses were performed using a Spearman’s test by GraphPad Prism Version 7 (GraphPad Software, La Jolla, CA, USA). For graphical illustration of data and statistical analyses between ctrl and *Nedd4-2^−/−^* after 12 weeks (*t*-test), GraphPad Prism version 5.0.1 (GraphPad Software, La Jolla, CA, USA) was used.

## Figures and Tables

**Figure 1 ijms-22-07607-f001:**
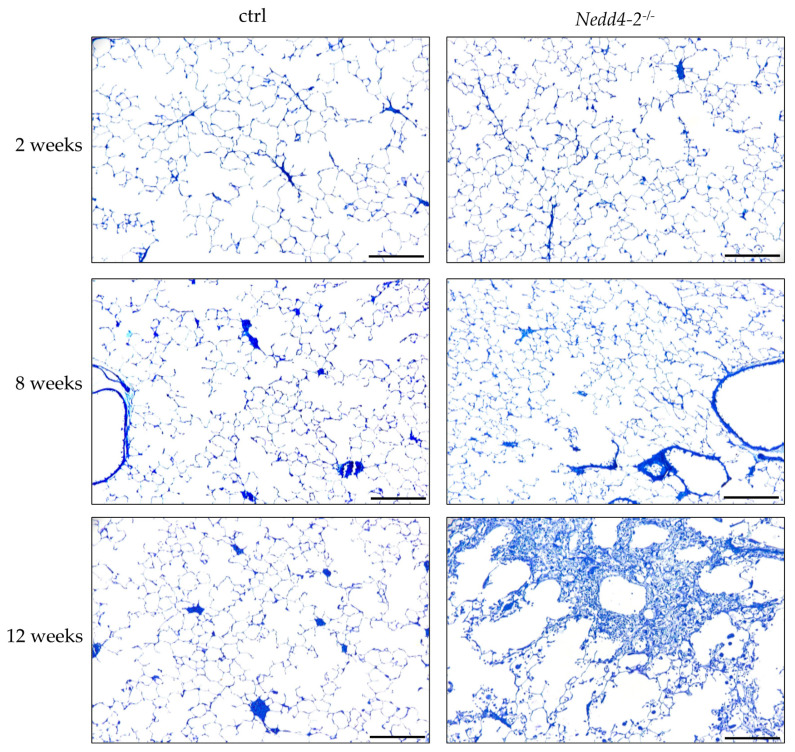
Composition of parenchyma. Representative light microscopic images of control (ctrl) and *Nedd4-2*-deficient lungs (*Nedd4-2*^−/−^) after 2, 8 and 12 weeks of doxycycline induction. At 12 weeks, septal walls appears thicker, and in some areas, lungs are consolidated by inflammatory infiltrates and fibrosis without recognizable acinar microarchitecture. Scale bar 200 µm.

**Figure 2 ijms-22-07607-f002:**
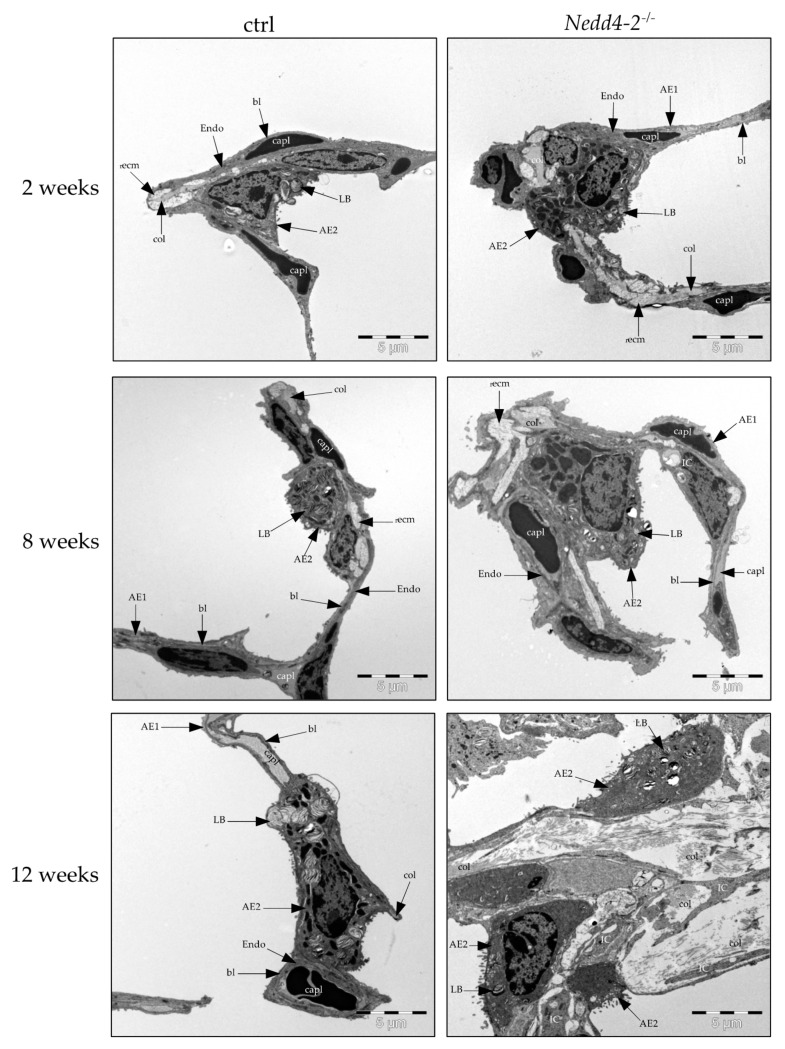
Composition of septal walls. Electron microscopic images of the ultrastructure of control (ctrl) and *Nedd4-2* deficient lungs (*Nedd4-2*^−/−^) after 2, 8 and 12 weeks of doxycycline induction. Increase of collagen fibrils, interstitial cells and alveolar epithelial type II (AE2) cells can be observed after 12 weeks. Abbreviations: AE1: alveolar epithelial type I cells; AE2: alveolar epithelial type II cells; LB: lamellar body; Endo: endothelial cells; IC: interstitial cells; _r_ecm: extracellular matrix without collagen fibrils; col: collagen fibrils; capl: capillary lumen (with erythrocyte); bl: basal lamina.

**Figure 3 ijms-22-07607-f003:**
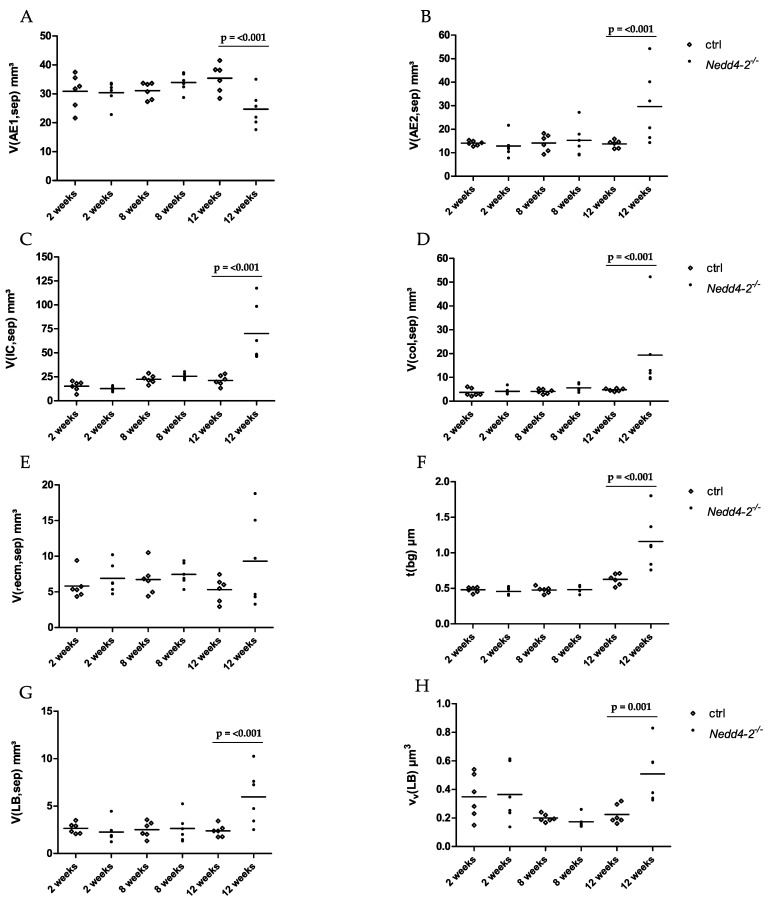
Septal microarchitecture of controls (ctrl) and *Nedd4-2*-deficient (*Nedd4-2*^−/−^) lungs. Stereological data determined at the electron microscopic level are provided in (**A**–**H**) and show total volume of AE1 cells within septal walls (**A**), total volume of AE2 cells within septal walls (**B**), total volume of interstitial cells (IC) within septal walls (**C**), total volume of collagen fibrils within septal walls (**D**), total volume of extracellular matrix without collagen fibrils within septal walls (**E**), arithmetic mean thickness of the blood–gas barrier, τ(bg) (**F**), total volume of lamellar bodies (LB) within septal walls (**G**) and volume-weighted mean volume of the lamellar bodies, ν_V_(LB) (**H**). Individual data and means are presented. The increase in the arithmetic mean thickness of the blood–gas barrier (**F**) after 12 weeks of *Nedd4-2* deficiency is predominantly attributable to an increase of the volume of AE2 cells (**B**), the volume of interstitial cells (**C**) and the volume of collagen fibrils (**D**) by simultaneous decrease of the volume of AE1 cells (**A**). In addition, there is an increase of the volume of lamellar bodies (**G**) and the volume-weighted mean volume of the lamellar bodies (**H**) after 12 weeks of *Nedd4-2* deficiency. Statistical analyses were based on a two-way ANOVA taking the factors genetics (G: ctrl vs. *Nedd4-2*^−/−^) and time (7: weeks) into account. Statistically significant differences after adjustment of the *p*-level for multiple testing using Tukey’s correction.

**Figure 4 ijms-22-07607-f004:**
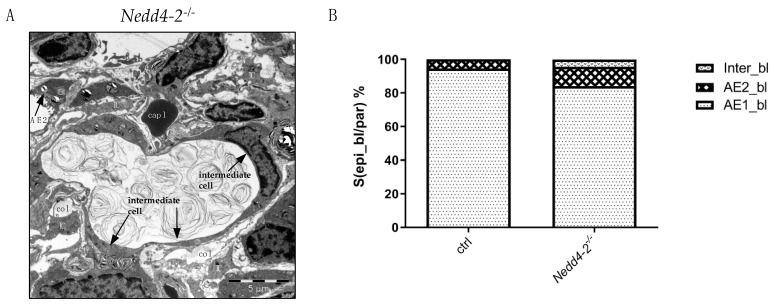
Composition of the alveolar epithelium after 12 weeks. (**A**) Electron microscopic image of the ultrastructure of intermediate cells and AE2 cells in *Nedd4-2*-deficient lungs (*Nedd4-2*^−/−^) after 12 weeks. Intermediate cells were only found after 12 weeks of *Nedd4-2* deficiency. These cells contain a few lamellar bodies and form extensions similar to squamous epithelial cells so that they have properties of both AE1 and AE2 cells. Abbreviations: AE2: alveolar epithelial type II cells; IC: interstitial cells; col: collagen fibrils. (**B**) Surface fraction of the basal lamina (bl) covered by AE1 cells (AE1_bl), AE2 cells (AE2_bl) and intermediate cells (Inter_bl) in controls (ctrl) and *Nedd4-2*-deficient (*Nedd4-2*^−/−^) lungs. There is a reduced surface area of the epithelial basal lamina covered by AE1 cells and a significantly increased surface area of the epithelial basal lamina covered by AE2 cells and intermediate cells. Data are provided as mean.

**Figure 5 ijms-22-07607-f005:**
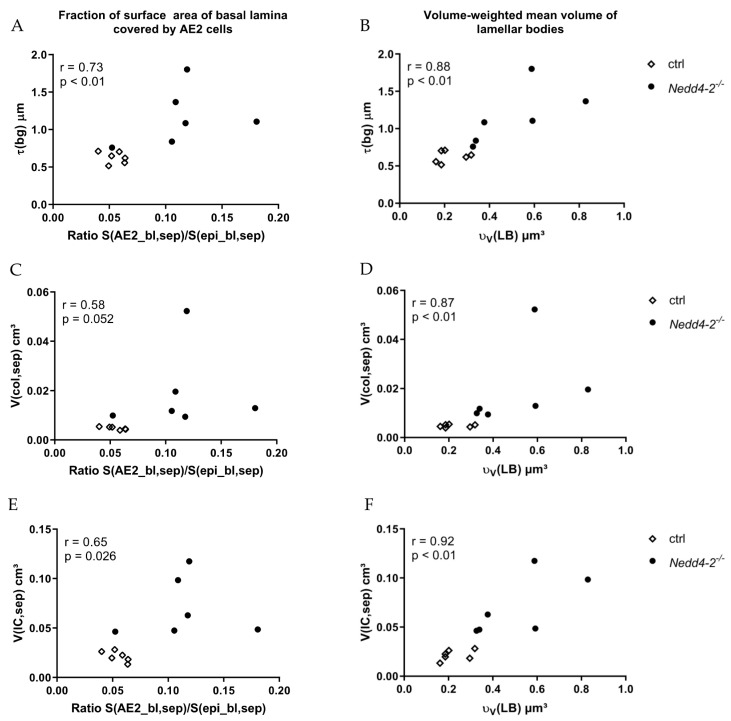
Correlation between different structural data of *Nedd4-2*-deficient (*Nedd4-2*^−/−^) lungs after 12 weeks. (**A**) The arithmetic mean thickness of the blood–gas barrier demonstrates a highly significant positive correlation with the fraction of surface area of basal lamina covered by AE2 cells and (**B**) a highly significant positive correlation with the volume-weighted mean volume of lamellar bodies. (**C**) The volume of collagen fibrils demonstrates no significant correlation with the fraction of surface area of basal lamina covered by AE2 cells and (**D**) a positive, highly significant correlation with the volume-weighted mean volume of lamellar bodies. (**E**) The volume of interstitial cells demonstrates a positive significant correlation with the fraction of surface area of basal lamina covered by AE2 cells and (**F**) a positive, highly significant correlation with the volume-weighted mean volume of lamellar bodies. The Spearman correlation coefficient r and statistical significance *p* are provided.

**Table 1 ijms-22-07607-t001:** Lung architecture.

Group	2 Weeks		8 Weeks		12 Weeks		Two-Way ANOVA		
Parameter	ctrl	*Nedd4-2^−/−^*	ctrl	*Nedd4-2* ^−/−^	ctrl	*Nedd4-2* ^−/−^	G	T	Inter
Females/males	4/2	3/3	3/3	3/3	4/2	3/3			
V(lung) cm³	0.59 (0.03)	0.60 (0.05)	0.64 (0.04)	0.61 (0.05)	0.80 (0.03)	0.85 (0.06)	n.s.	<0.001	0.028
V(par,lung) cm³	0.51 (0.04)	0.51 (0.04)	0.54 (0.03)	0.51 (0.04)	0.65 (0.03)	0.68 (0.07)	n.s.	<0.001	n.s.
V(sep,par) cm³	0.13 (0.02)	0.12 (0.01)	0.13 (0.01)	0.16 (0.01)	0.16 (0.02)	0.22 (0.05)	0.001	<0.001	0.011
V(air,par) cm³	0.38 (0.05)	0.39 (0.03)	0.40 (0.02)	0.36 (0.04)	0.49 (0.02)	0.40 (0.09)	0.016	0.004	n.s.
V(cons,lung) cm³	0.00 (0.00)	0.00 (0.00)	0.00 (0.00)	0.00 (0.00)	0.00 (0.00)	0.05 (0.03)	**–––**	**–––**	<0.001

Summarized light microscopic data are provided as mean (standard deviation). Parameters are defined in the text and in the methods section. Statistical analysis was based on a two-way ANOVA taking the factors genetics (G: ctrl vs. *Nedd4-2*^−/−^) and time (T: weeks) as well as the interaction (Inter) of these factors into account. *p*-values below 0.05 were considered as statistically significant. Abbreviations: V: volume; par: lung parenchyma; sep: interalveolar septa; air: acinar airspaces; cons: consolidated lung parenchyma.

**Table 2 ijms-22-07607-t002:** Composition of interalveolar septa.

Group	2 Weeks		8 Weeks		12 Weeks		Two-Way ANOVA		
Parameter	ctrl	*Nedd4-2* ^−/−^	ctrl	*Nedd4-2* ^−/−^	ctrl	*Nedd4-2* ^−/−^	G	T	Inter
V(AE1,sep) mm³	30.88 (5.96)	30.43 (4.02)	31.16 (2.89)	33.93 (3.15)	35.40 (4.92)	24.73 (6.24)	n.s.	0.004	n.s.
V(AE2,sep) mm³	14.07 (0.98)	12.85 (4.71)	14.19 (3.60)	15.29 (6.74)	13.75 (1.62)	29.64 (15.59)	n.s.	0.014	0.020
V(Endo,sep) mm³	36.78 (10.80)	34.80 (6.63)	34.50 (4.83)	46.72 (3.35)	42.61 (10.07)	35.88 (8.54)	n.s.	0.016	n.s.
V(capl,sep) mm³	21.49 (2.50)	23.05 (5.61)	20.72 (3.92)	24.18 (5.21)	35.42 (5.74)	26.94 (6.98)	n.s.	0.018	<0.001
V(_r_ecm,sep) mm³	5.81 (1.84)	6.89 (2.10)	6.75 (2.15)	7.46 (1.53)	5.33 (1.70)	9.30 (6.43)	n.s.	n.s.	n.s.
V(col,sep) mm³	3.68 (1.64)	4.13 (1.42)	4.10 (0.98)	5.57 (1.59)	4.76 (0.60)	19.33 (16.56)	<0.001	<0.001	0.004
V(IC,sep) mm³	15.29 (5.14)	12.78 (2.67)	22.58 (4.42)	25.67 (3.11)	21.35 (5.44)	70.19 (30.47)	<0.001	<0.001	<0.001
V(LB,sep) mm³	2.65 (0.57)	2.27 (1.14)	2.52 (0.84)	2.63 (1.45)	2.40 (0.63)	5.97 (2.92)	0.034	0.013	0.005
ν_V_(LB) µm^3^	0.35 (0.16)	0.36 (0.20)	0.20 (0.03)	0.17 (0.04)	0.22 (0.07)	0.51 (0.20)	n.s.	0.004	0.018
S(epi_air,sep) cm²	1322 (181)	1305 (205)	1386 (210)	1622 (150)	2232 (191)	1518 (501)	<0.001	n.s.	<0.001
τ(bg) µm	0.48 (0.04)	0.46 (0.05)	0.48 (0.05)	0.48 (0.05)	0.83 (0.10)	1.29 (0.41)	0.0027	<0.001	0.002

Summarized electron microscopic data are provided as mean (standard deviation). Parameters are defined in the text and in the methods section. Statistical analysis was based on a two-way ANOVA taking the factors genetics (G: ctrl vs. *Nedd4-2*^−/−^) and time (T: weeks) as well as the interaction (Inter) of these factors into account. *p*-values below 0.05 were considered as statistically significant. Abbreviations: V: volume; ν_V_: volume-weighted mean volume; S: surface area; τ: arithmetic mean thickness; AE1: alveolar epithelial type I cell; AE2: alveolar epithelial type II cell; Endo: endothelium; _r_ecm: residual extracellular matrix; col: collagen fibrils; IC: interstitial cell; LB: lamellar body; epi_air: epithelium covered by air; bg: blood–gas barrier; sep: interalveolar septa.

**Table 3 ijms-22-07607-t003:** Surface areas of epithelial cells after 12 weeks.

Group	12 Weeks		
Parameter	Ctrl	*Nedd4-2* ^−/−^	*t*-Test
S(AE1_bl,sep) cm²	1702 (156)	1221 (370)	0.015
S(AE2_bl,sep) cm²	98.0 (16.7)	160.8 (60.5)	0.034
S(Inter_bl,sep) cm²	0.00 (0.00)	59.6 (16.8)	<0.01
S(epi_bl,sep) cm²	1799 (159]	1442 (381)	0.060
S(AE1_air,par) cm²	2085 (174]	1302 (418)	<0.01
S(AE2_air,par) cm²	147 (27.1)	216 (119)	n.s.
S(Inter_air,par) cm²	0.00 (0.00)	67.1 (25.1)	<0.01
S(epi_air,par) cm²	2232 (501)	1518 [191)	<0.01
S(endo,sep) cm²	1707 (203)	1963 (370)	n.s.

Electron microscopic data after 12 weeks are provided as mean (standard deviation). Parameters are defined in the text and in the methods section. Statistical analyses were based on a *t*-test taking the factor genetics (G: ctrl vs. *Nedd4-2*^−/−^) into account. *p*-levels below 0.05 were considered as statistically significant, n.s.: not significant. Abbreviations: S: surface area; AE1_bl: basal lamina covered by alveolar epithelial type I cells; AE2_bl: basal lamina covered by alveolar epithelial type II cells; Inter_bl: basal lamina covered by intermediate alveolar epithelial cells; AE1_air: alveolar epithelial type I cells covered by air; AE2_air: alveolar epithelial type II cells covered by air; Inter_air: intermediate alveolar epithelial cells covered by air; epi_air: alveolar epithelium covered by air; endo: endothelial cells.

**Table 4 ijms-22-07607-t004:** Definition of light microscopic stereological parameters.

Stereological Parameters (Light Microscopy)		
Parameter	Test System	Primary Magnification
Volume of parenchyma, V(par,lung)	Point counting	5×
Volume of non-parenchyma, V(non-par,lung)	Point counting	5×
Volume of (ductal and alveoli) airspace, V(air,par)	Point counting	20×
Volume of septal wall, V(sep,par)	Point counting	20×
Volume of consolidated, destroyed tissue areas, V(cons,lung)	Point counting	20×

**Table 5 ijms-22-07607-t005:** Definition of electron microscopic stereological parameters.

Stereological Parameters (Electron Microscopy)		
Parameter	Test System	Primary Magnification
Volume of alveolar epithelial type 1 cells in septal walls, V(AE1,sep)	Point counting	8900×
Volume of alveolar epithelial type 2 cells in septal walls, V(AE2,sep)	Point counting	8900×
Volume of endothelial cells within septal walls, V(Endo,sep)	Point counting	8900×
Volume of interstitial cells within septal walls, V(IC,sep)	Point counting	8900×
Volume of collagen fibrils within septal walls, V(col,sep)	Point counting	8900×
Volume of the extracellular matrix without collagen fibrils within septal walls, V(_r_ecm,sep)	Point counting	8900×
Volume of capillary lumen within septal walls, V(capl,sep)	Point counting	8900×
Surface area of endothelium within septal walls, S(endo,sep)	Point and intersection counting	8900×
Surface area of of basal lamina covered by alveolar epithelium within septal walls, S(epi_bl,sep) (12 weeks)	Point and intersection counting	8900×
Surface area of the basal lamina covered by AE1, S(AE1_bl,sep) (12 weeks)	Intersection counting	8900×
Surface area of the basal lamina covered by AE2, S(AE2_bl,sep) (12 weeks)	Intersection counting	8900×
Surface area of the basal lamina covered by intermediate cells, S(Inter_bl,sep) (12 weeks)	Intersection counting	8900×
Surface area of alveolar epithelium covered by air within parenchyma, S(epi_air,par)	Intersection counting	8900×
Surface area of the AE1 covered by air within parenchyma, S(AE1_air,par) (12 weeks)	Intersection counting	8900×
Surface area of the AE2 covererd by air within parenchyma, S(AE2_air,par) (12weeks)	Intersection counting	8900×
Surface area of the intermediate cells covered by air within parenchyma, S(Inter_air, par) (12 weeks)	Intersection counting	8900×
Arithmetic mean thickness of the blood–gas barrier, τ(bg)	Volume-to-surface ratio	8900×
Volume of lamellar bodies within the septal walls V(LB,sep)	Point counting	7100×
Volume-weighted mean volume of the LB’s, ν_v_(LB)	Point sampling intercept method	7100×

## Data Availability

The data presented in this study are available upon request from the corresponding author.

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
