# Peer review of "Linking Fibrotic Remodeling and Ultrastructural Alterations of Alveolar Epithelial Cells after Deletion of Nedd4-2"

_ijms, 2021, doi:10.3390/ijms22147607_

Round 1

Reviewer 1 Report

The present manuscript by Engelmann et al. seeks to characterize the time course of a progressive pulmonary fibrosis in animals lacking the ubiquitin ligase Nedd4-2 in type II pneumocytes and club cells (with deletion induced by doxycycline administration when animals are 4-to-6 weeks of age) using design-based stereological investigation of light and electron microscopy.  The stated objective of this work was to determine whether changes in the alveolar epithelium precede, correlate with, or are a consequence of the observed interstitial remodeling.  However, from the results obtained, it seems that an inadequate number of timepoints were selected to draw any rigorous conclusions about this hypothesis.  This leaves the remainder of the findings standing as descriptive without much additional significant insight to the field.  In order to make this report worthy of publication, the following points should be addressed:

Major Points:

 1) As there is a previously reported "dramatic acceleration" in the deterioration of the lung structure between 8 and 12 weeks following doxycycline induction in this model, it seems that before any conclusions can be made, an intermediary timepoint of 10 weeks post induction is needed.  This is particularly relevant given the fact that fibrotic changes could be observed within 2 weeks of amiodarone treatment or adenoviral-mediated delivery of activated TGF-beta1 to lung epithelium (as the authors point out in the discussion).  Thus, a lot can happen in two weeks.  This might also allow for the determination of whether or not the alveolar epithelium is a driver of the pathophysiological changes associated with interstitial lung remodeling since there could be pioneer changes visible at 10 weeks post doxycyline treatment that were not present at 8 weeks.  It is also possible that consolidation might be less of a factor at this particular timepoint and more theoretical "precursor lesions" might be observed in those areas destined to undergo consolidation.

2) Due to known gender differences in IPF, the authors should report the gender of the animals used in their experiments to ensure that any potential modifying effects of sex are not confounding the interpretation of the results.

3) As Wnt/beta-catenin signaling has been implicated in the inhibition of transdifferentiation, it would bolster the author's speculations to assess whether there is any change in this signaling pathway in all the timepoints examined.

Minor Points:

1) There is some editing that needs to be conducted on this manuscript as there are some awkward word choices that likely results from translation to English (e.g., "destructed" rather than "destroyed").

2) The authors should clearly define the "Inter" heading in those tables that present 2-way ANOVA analysis (Tables 1&2).

Author Response

We thank the reviewers for the very valuable and contructive feddback! 

Reviewer 2 Report

This manuscript by Dr Engelmann and colleagues describes the structural alteration within the lung of a mouse model of lung fibrosis. The study is a follow-up on a previous study on this model, published in 2020. The authors use a stereologic approach to assess structural differences in the lungs of these mice at 3 different time-point, 2, 8 and 12 weeks after initiation of the doxycycline to knock out nedd4-2. Overall, this is an excellent study; I only have relatively minor and very generic comments.

1) The introduction could be improved by providing the hypothesis that is being tested.

2) Some of the data is presented twice, in a table and in a scatter-plot. This redundancy should be removed. I personally prefer the scatter plots.

3) The discussion is too long and could be cut by approximately 30%. Specifically, information that is solely based on the previous studies without (or with weak) linkage to the current data set can be removed.

4) The authors use the term “predate” frequently, in the context used in this study, precede may be a more appropriate term.

5) Overall, the data obtained at 2 weeks appear to be somewhat irrelevant for the conclusion.  This data could either be removed or its relevance/inclusion should be better rationalized.

Author Response

We thank the reviewers for the very valuable and constructive feedback!

Round 2

Reviewer 1 Report

I think the authors have adequately addressed my concerns with the present revision.  The only remaining issue before publication would be thorough vetting of the English language and style.  The following is a small list of some stylistic errors but is in no way comprehensive of edits that need to be made throughout the manuscript:

1) Be sure that proper gene conventions are followed.  In particular, gene names should always be italicized (e.g., in intro, MUC5B gain of function variants should be italicized).  Further, the official mouse gene symbol for Nedd4-2 is actually Nedd4l (human = NEDD4L).  It would be useful to acknowledge this on the first usage if the colloquial Nedd4-2 is to be used.

2) I believe I poorly explained in my previous review -- "destroyed" is the proper adjective rather than "destructed." 

3) In the newly described central questions outlined in the introduction, question 1 would be better phrased as "Does the conditional deletion of Nedd4-2 have a direct effect on the..."  Further, for question 3, better phrasing would be "In this animal model of progressive pulmonary fibrosis, is there any evidence..."

4) line 123: Try something like, "This study was based on our previous findings that demonstrated the absence of light microscopic abnormalities at 2 weeks post induction of Nedd4-2 deficiency but a discrete thickening of septal...at 8-12 weeks after Nedd4-2 deletion."

Author Response

We thank the reviewer for this very constructive feedback!

General comment: I think the authors have adequately addressed my concerns with the present revision.  The only remaining issue before publication would be thorough vetting of the English language and style.  The following is a small list of some stylistic errors but is in no way comprehensive of edits that need to be made throughout the manuscript:

Reply: We have added another round of proof-reading and could identify several errors which were corrected in the revised version.

Comment 1: Be sure that proper gene conventions are followed.  In particular, gene names should always be italicized (e.g., in intro, MUC5B gain of function variants should be italicized).  Further, the official mouse gene symbol for Nedd4-2 is actually Nedd4l (human = NEDD4L).  It would be useful to acknowledge this on the first usage if the colloquial Nedd4-2 is to be used.

Reply 1: We have adjusted the style of the gene names according to the convention.

Comment 2: I believe I poorly explained in my previous review -- "destroyed" is the proper adjective rather than "destructed."

Reply 2: We have replaced the term “destructed” by the term “destroyed” as suggested.

Comment 3: In the newly described central questions outlined in the introduction, question 1 would be better phrased as "Does the conditional deletion of Nedd4-2 have a direct effect on the..."  Further, for question 3, better phrasing would be "In this animal model of progressive pulmonary fibrosis, is there any evidence..."

Reply 3: Thank you very much for this suggestion! We have adopted the research questions accordingly.

Comment 4: line 123: Try something like, "This study was based on our previous findings that demonstrated the absence of light microscopic abnormalities at 2 weeks post induction of Nedd4-2 deficiency but a discrete thickening of septal...at 8-12 weeks after Nedd4-2 deletion."

Reply 4: Thank you very much! We have rewritten this sentence as suggested.